# Invasive Species Appearance and Climate Change Correspond with Dramatic Regime Shift in Thermal Guild Composition of Lake Huron Beach Fish Assemblages

Jessica Bowser [1,2], Tracy Galarowicz [1], Brent Murry [3,*] and Jim Johnson [4,†]

1 Department of Biology, Central Michigan University, Biosciences 2100, Mount Pleasant, MI 48859, USA
2 Alpena Fish and Wildlife Conservation Office–Detroit River Substation, John D. Dingell Visitor Center, 5437 West Jefferson Ave, Trenton, MI 48183, USA
3 Division of Forest and Natural Resources, Davis College, West Virginia University, 322 Percival Hall, 1145 Evansdale Dr., Morgantown, WV 26506, USA
4 Michigan Department of Natural Resources, 160 E. Fletcher St, Alpena, MI 49707, USA
* Correspondence: brent.murry@mail.wvu.edu
† Retired.

**Abstract:** Lake Huron has undergone dramatic and well-documented lake-wide food web changes as a result of non-native species introductions. Coastal beaches, which serve as nursery habitats for native and introduced species, are, however, relatively poorly studied. Our objective was to assess fish assemblages of beach habitats in western Lake Huron and compare species composition pre- (1993) and post-invasion (2012) of dreissenid mussels and round goby (*Neogobius melanostomus*). Nearshore beach fish assemblages were sampled by nighttime beach seining during spring and summer in 1993 and 2012 in the western basin of Lake Huron along the Michigan shoreline. Catch rates were considerably higher, but there were fewer species present in 2012 than in 1993. The composition of species changed dramatically from a cold- and cool-water species assemblage in 1993 (dominated by alewife (*Alosa pseudoharengus*), spottail shiner (*Notropis hudsonius*), and lake whitefish (*Coregonus clupeaformis*), as well as Chinook salmon (*Oncorhynchus tshawytscha*) and rainbow smelt (*Osmerus mordax*)) to a cool- and warm-water species assemblage in 2012 (dominated by cyprinids, round goby (*Nogobius melanstomus*), and yellow perch (*Perca flavescens*)). The observed rise in catch rate and shift in species composition appears related to the introduction of invasive species as well as an on-going warming pattern in nearshore waters.

**Keywords:** regime shift; Lake Huron; beach habitat; fish assemblages; invasive species; climate change





## 1. Introduction

Fish assemblages are shaped by species interactions, food availability, habitat, and other abiotic and biotic factors [1–4]. Competition and predation along with other biotic interactions can impact fish species found in freshwater systems and how these systems are structured [1]. The effects of these interactions can be further influenced by abiotic factors such as temperature, including seasonal variation, and substrate [5,6]. Both biotic interactions and abiotic variables in the Great Lakes system have the ability to influence and alter the structure of fish assemblages [7,8].

Fish assemblages in the Great Lakes have been affected by the introduction of non-native and invasive species resulting in the loss or decline of native species [9–11]. Unintentional and intentional introductions of non-native species in Lake Huron have forever affected the lake's food web structure [9,12]. Although invasion dates in Lake Huron are not well documented, the zebra mussel (*Dreissena polymorpha*) and the quagga mussel (*D. bugensis*), originating from the Ponto-Caspian Sea region, were introduced by ship ballast water discharge into the Great Lakes in the late 1980s [11,13]. Shortly after, the

round goby (*Neogobius melanostomus*), also from the Ponto-Caspian region and transported via ballast water discharge, invaded the Great Lakes. The round goby was first reported in Lake Huron in 1994 in Goderich, Ontario [13].

The introduction of dreissenid mussels facilitated the success of the round goby, providing a food source from its native range [14,15]. Dreissenid mussels are also partially responsible for a major ecological change in energy flow, often referred to as a "shunt", where pelagic phosphorus is transferred to the benthic environment. The transfer drastically alters energy pathways, resulting in phosphorus enrichment in the littoral zone [11,16–19]. Consumption of *Dreissena* spp. by round goby transfers this stored energy from the benthos to predator fish [18,20]. The round goby is now considered an important prey species to many fish, including smallmouth bass (*Micropterus dolomieu*), walleye (*Sander vitreus)*, and lake trout (*Salvelinus namaycush*) [18–23].

By 2003, the fish assemblage in Lake Huron changed again. Non-native alewife (*Alosa pseudoharengus*) populations declined [10,11,22,24] concurrent with their main prey source, *Diporeia* spp. [7,19,25]. Stocking efforts of non-native salmonids were reduced, and their populations began to decline as well [6,26,27]. Population numbers of native species such as walleye and lake trout have rebounded concurrent with the decline in alewife [11,13,28,29]. Invasive species and anthropogenic influences have both short- and long-term impacts, but it is worth noting that recent changes in Lake Huron fish assemblages have occurred over only a few decades [10].

Changes in water temperature also have the capacity to alter species assembly, especially under climate change in both fresh [30] and marine waters [31,32]. The Great Lakes have experienced a steady increase in water temperatures over the last couple of decades [30,33], but it is unknown whether the lakes have surpassed a critical threshold driving changes in species composition.

Nearshore and coastal habitats are particularly susceptible to invasive species and temperature changes. Beaches make up around 20% of all the shoreline in the Great Lakes [34]. However, when compared to other habitat research topics, beaches are tremendously understudied [34]. Therefore, observing nearshore fish assemblages is essential given past and current changes to the Lake Huron ecosystem and management efforts to recover economically significant fisheries.

The goal of this study was to compare and contrast the nearshore beach fish assemblages sampled in 1993 and 2012 in western Lake Huron. We related changes in the beach fish assemblage to increases in the number and dominance of non-native fish species and warming spring and early summer water temperatures. A seining survey was conducted in 1993 by the Michigan Department of Natural Resources and the U. S. Fish and Wildlife Service to quantify natural recruitment and nursery areas of Chinook salmon (*Oncorhynchus tshawytscha*) [35]. The most abundant species were alewife, spottail shiner (*Notropis hudsonius*), and sand shiner (*Notropis stramineus*) [35]. We hypothesized that fish diversity would increase during the more recent sampling associated with the transition from a largely alewife and salmonid (cold- and cool-water species) dominated system to a goby and percid (cool and warm water species) dominated system due to recent changes in the Lake Huron food web from 1993 to 2012. We further hypothesized that these changes were related to both the introduction of non-native species as well as increasing water temperatures.

## 2. Methods

Nearshore beach fish assemblages in Lake Huron were sampled at seven sites in 1993 and six sites in 2012 along the Michigan shoreline (Figure 1). Beach habitat across western Lake Huron is generally characterized by extremely shallow slopes where water depths are frequently under a meter several hundred meters away from the shoreline. The substrate is generally moderately packed, but highly mobile sand with very little submersed cover (e.g., limited isolated small rocks, very little aquatic vegetation, very infrequent woody debris). The shallow water tends to warm rapidly but can be highly variable due to exchange with offshore deeper waters during regular seiche events.

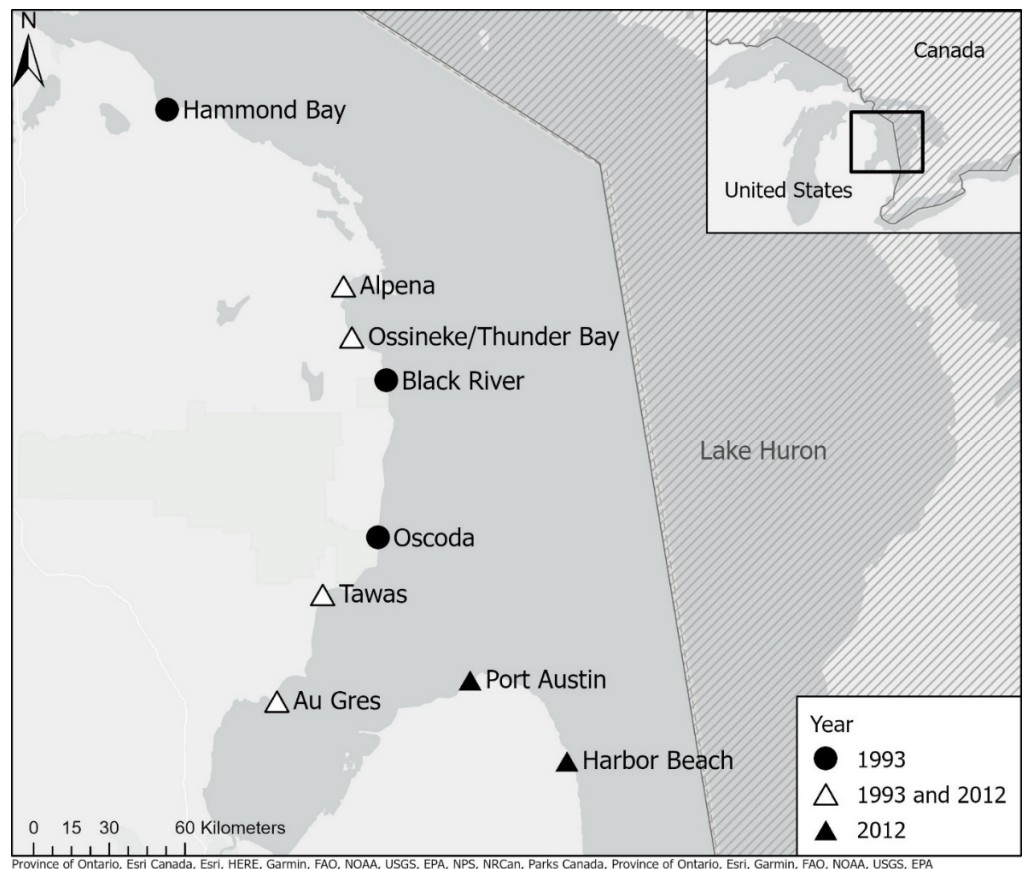

**Figure 1.** Locations of study sites along the Michigan shoreline of the United States in western Lake Huron sampled in 1993 [35] and 2012.

The 1993 data collection was a joint operation by Michigan Department of Natural Resources and U.S. Fish and Wildlife Service [35], and data were available through one of the original project members and coauthor of this manuscript (James Johnson). Sampling during 2012 followed the same methods, used the same equipment, and sampled a similar geography across western Lake Huron. In 2012, we sampled four of the exact same beaches as in 1993, and we did not sample the northernmost site sampled in 1993, nor two of the regionally central sites, but added two more eastern and southern sites (Figure 1). We were striving to understand beach fish assemblages of the region more so than at specific fixed sites, we feel that the collection of sites from both years are equally representative of the region as a whole. Fish were sampled by beach seining following procedures in [35]. A 47.5 m seine (15.2 m bunt with 9.5 mm mesh and 12.7 mm wings) was pulled perpendicular to the shoreline the length of the seine, then walked in a parabola back to shore approximately 5 m from the originating point. Sampling occurred weekly at all sites during spring (May 15–June 27; Week 1: May 14–20, Week 2: May 21–27, Week 3: May 28–31, Week 4: June 4–5, Week 5: June 11–13, Week 6: June 19–21, and Week 7: June 25–27) at least an hour after sunset. The temporal sampling regime was consistent across both sampling periods (1993 and 2012). Two replicate seine hauls were performed at each site on each sampling date. Fish greater than 150 mm were identified to the lowest taxonomic group possible, measured (total length, mm), and released. All other fish were euthanized in MS-222 (Tricaine methane sulfonate, U.S. Food and Drug Administration approved fish anesthetic), preserved in 5% formalin, and identified in the laboratory.

Patterns in assemblage composition between time periods (1993 and 2012) were examined with Nonmetric Multidimensional Scaling (NMDS) and Multi-Response Permutation Procedure (MRPP). Optimal solutions (e.g., the number of axes) were determined through low stress values and Monte Carlo randomization tests (*n* = 100 runs) [36]. Distance cal-

culations and ordinations were performed with PC-ORD version 6. We used MRPP to test the null hypotheses: no assemblage differences in species composition between time periods. Euclidean (Pythagorean) distances were calculated for untransformed abundance data and with replicated seine hauls pooled in each analysis. The MRPP gives two statistics: an agreement statistic (*A*) describing the degree of within-group homogeneity compared to that expected by chance (e.g., effect size), and a *p*-value that estimates the probability that observed differences are due to chance. Values for *A* statistic indicate within-group heterogeneity (*A* = 1), level of heterogeneity within groups is equal to what is expected by chance (*A* = 0), or more heterogeneity within groups than expected by chance (*A* < 0). For ecological communities, *A* values are typically <0.1 and values ≥0.3 suggest differences between assemblages [36]. Pearson and Kendall correlations were calculated for each axis to determine correlations between the axis and individual species. Indicator species analyses were also performed to describe the value of different species in the different time periods. Indicator values (IV) from the indicator species analysis range from 0 with no indication to 100 with perfect indication [36].

Total catch per unit effort (CPUE) was calculated as the number of fish per seine haul (#/seine haul) for each sampling event. Species that both represented at least 1% of the fish collected and were present at four or more of the sites were considered dominant species. Species that did not meet these parameters were grouped and summarized by family. If the family grouping did not represent at least 1% of the total number of fish collected and were not present at four or more of the sites, they were grouped into an "other" category. CPUE of the dominant species and families were compared between time periods with a repeated measures analysis of variance (ANOVA). All within-subject factors were checked for sphericity using Mauchly's test of sphericity. If the within-subject factors failed to meet the assumptions of sphericity, the Huynh–Feldt correction factor for adjusted *p*-values was used. CPUE data were $\log_{10}+1$ transformed to meet the assumptions of normality.

Proportions of the catch by each group (dominant species or family) by number were calculated for each sampling event. The differences in proportion of each dominant species were examined with repeated measures ANOVA. All within-subject factors were checked for sphericity using Mauchly's test of sphericity. If the within-subject factors failed to meet the assumptions of sphericity, the Huynh–Feldt correction factor for adjusted *p* values was used. Proportion data were arcsine transformed to meet the assumptions of normality. All statistical analyses were performed in SAS (GLM procedure, version 9.4).

The influence of water temperature on the species composition was examined in both regional and local contexts and compared to the thermal tolerance data for specific species. Regionally, we downloaded water temperature (°C) data from the NOAA buoy #45008 in South Lake Huron at 43 NM East of Oscoda, MI (44.283 N 82.416 W; https://www.ndbc.noaa.gov/station_history.php?station=45008, accessed on 11 November 2019) for the months of May and June from 1993 through 2012. This buoy was selected because it was closest and in the center of the field sites sampled. We examined the annual monthly means across the time series with linear regression to test the null hypothesis of no increase in mean monthly temperature through the time period (PROC REG, SAS, version 9.4). Water temperature (°C) was also recorded locally during each sampling event in 1993 and 2012. Increases in water temperature during the sampling season within a year were examined with linear regression (PROC REG, SAS, version 9.4). Temporal differences among years were tested by comparing slopes of the daily water temperature data using year as a categorical covariate (PROC MIXED, SAS, version 9.4). We also compiled thermal guild, preferred temperatures, and/or final temperature preferendum of the dominant fish species from the literature [37,38] to better understand changes in fish assemblages.

## 3. Results

*Fish Abundance*

In 1993, 17,206 fish were collected representing 47 species in 96 seine hauls [35]. We captured 37,876 fish in 2012 representing 29 species in 72 seine hauls. CPUE of all species was higher in 2012 (mean = 526.1 fish/seine haul, SE = 116.4) relative to 1993 (mean = 179.3 fish/seine haul, SE = 24.9), and there was a significant interaction between week and year (Year: $F_{1,14}$ = 16.36, $p$ < 0.01; Week: $F_{6,84}$ = 0.88, $p$ = 0.51; Year*Week: $F_{6,84}$ = 2.46, $p$ = 0.03. The CPUE of alewife, lake whitefish, longnose dace, rainbow smelt, spottail shiner, trout perch, cyprinidae, and salmonidae were higher in 1993 than in 2012 (Table 1, Figure 2A–Q). The CPUE of bluntnose minnow, emerald shiner, mimic shiner, round goby, white perch, catostomidae, centrarchidae, and ictaluridae were higher in 2012 than 1993. A significant interaction existed between week and year for common shiner, emerald shiner, longnose dace, round goby, centrarchidae, and ictaluridae (Table 1, Figure 2A–Q).

**Table 1.** Results of repeated measures analyses of variance (ANOVA) of catch per unit effort (CPUE) and proportion of catch by number of the dominant species or families of the nearshore beach fish assemblages between time periods (1993 vs. 2012). The Huynh–Feldt correction factor for adjusted $p$ values was used for all analyses unless indicated.

| Species or Family | Variable | *df* | CPUE | | Number | |
|---|---|---|---|---|---|---|
| | | | *F* | *p* | *F* | *p* |
| Alewife | Year | 1 | 7.21 | 0.04 | 4.32 | 0.05 |
| (*Alosa pseudoharengus*) | Week | 6 | 2.44 | 0.15 | 3.58 | 0.03 |
| | Week*Year | 6 | 1.63 | 0.25 | 3.76 | 0.57 |
| Banded killifish | Year | 1 | 4.56 | 0.06 | 2.2 | 0.15 |
| (*Fundulus diaphanous*) | Week | 6 | 4.7 | 0.053 | 3.69 | 0.02 |
| | Week*Year | 6 | 4.7 | 0.053 | 2.49 | 0.08 |
| Bluntnose minnow | Year | 1 | 208.82 | <0.0001 | 0.63 | 0.44 |
| (*Pimephales notatus*) | Week | 6 | 2.37 | 0.14 | 1.16 | 0.33 |
| | Week*Year | 6 | 2.37 | 0.14 | 3.8 | 0.01 |
| Chinook salmon | Year | 1 | 1.30 | 0.29 | 2.45 | 0.17 |
| (*Oncorhynchus tshawytscha*) | Week | 6 | 1.30 | 0.28 | 1.84 | 0.12 |
| | Week*Year | 6 | 1.30 | 0.28 | 1.84 | 0.12 |
| Common shiner | Year | 1 | 3.99 | 0.08 | 21.83 | 0.00 |
| (*Luxilus cornutus*) | Week | 6 | 9.3 | 0.0002 | 0.53 | 0.63 |
| | Week*Year | 6 | 12.4 | <0.0001 | 0.49 | 0.66 |
| Emerald shiner | Year | 1 | 18.24 | 0.004 | 2.98 | 0.10 |
| (*Notropis atherinoides*) | Week | 6 | 9.46 | <0.0001 | 1.39 | 0.26 |
| | Week*Year | 6 | 18.38 | <0.0001 | 1.39 | 0.26 |
| Lake whitefish | Year | 1 | 3.5 | 0.008 | 2.13 | 0.16 |
| (*Coregonus clupeaformis*) | Week | 6 | 3.5 | 0.08 | 1.01 | 0.34 |
| | Week*Year | 6 | 3.5 | 0.08 | 1.01 | 0.33 |
| Longnose dace | Year | 1 | 359.44 | <0.0001 | 4.55 | 0.04 |
| (*Rhinichthys cataractae*) | Week | 6 | 177.66 | <0.0001 | 2.71 | 0.08 |
| | Week*Year | 6 | 179.89 | <0.0001 | 2.71 | 0.08 |
| Mimic shiner | Year | 1 | 25.61 | 0.0007 | 2.12 | 0.16 |
| (*Notropis volucellus*) | Week | 6 | 3.24 | 0.04 | 4.75 | 0.00 |
| | Week*Year | 6 | 2.63 | 0.08 | 1.02 | 0.39 |

**Table 1.** *Cont.*

| Species or Family | Variable | df | CPUE | | Number | |
|---|---|---|---|---|---|---|
| | | | F | p | F | p |
| Rainbow smelt | Year | 1 | 15.96 | 0.001 | 3.16 | 0.09 |
| (*Osmerus mordax*) | Week | 6 | 2.72 | 0.09 | 2.77 | 0.04 |
| | Week*Year | 6 | 2.72 | 0.09 | 9.18 | 0.00 |
| Round goby | Year | 1 | 59.54 | <0.0001 | 24.92 | 0.00 |
| (*Neogobius melanostomus*) | Week | 6 | 7.76 | 0.0005 | 5.9 | 0.002 |
| | Week*Year | 6 | 7.79 | 0.0005 | 9.24 | 0.00 |
| Sand shiner | Year | 1 | 0.58 | 0.48 | 1.34 | 0.26 |
| (*Notropis stramineus*) | Week | 6 | 1.71 | 0.15 | 0.85 | 0.42 |
| | Week*Year | 6 | 3.03 | 0.02 | 0.87 | 0.42 |
| Spotfin shiner | Year | 1 | 2.93 | 0.13 | 1.43 | 0.24 |
| (*Cyprinella spiloptera*) | Week | 6 | 1.29 | 0.29 | 0.93 | 0.35 |
| | Week*Year | 6 | 1.7 | 0.15 | 0.93 | 0.35 |
| Spottail shiner | Year | 1 | 788.38 | 0.001 | 9.28 | 0.01 |
| (*Notropis hudsonius*) | Week | 6 | 0.42 | 0.77 | 4 | 0.02 |
| | Year*Week | 6 | 0.25 | 0.88 | 3.63 | 0.02 |
| Trout perch | Year | 1 | 5.8 | 0.05 | 0.01 | 0.91 |
| (*Percopsis omiscomaycus*) | Week | 6 | 0.8 | 0.52 | 1.02 | 0.36 |
| | Year*Week | 6 | 2.17 | 0.11 | 3.86 | 0.03 |
| Yellow perch | Year | 1 | 3.91 | 0.08 | 1.51 | 0.24 |
| (*Perca flavescens*) | Week | 6 | 0.97 | 0.42 | 3.74 | 0.01 |
| | Year*Week | 6 | 0.82 | 0.49 | 0.54 | 0.69 |
| Catastomidae | Year | 1 | 7.21 | 0.044 | 4.44 | 0.05 |
| | Week | 6 | 0.05 | 0.15 | 1.07 | 0.38 |
| | Year*Week | 6 | 0.17 | 0.25 | 1.47 | 0.23 |
| Centrarchidae | Year | 1 | 15.71 | 0.004 | 1.19 | 0.29 |
| | Week | 6 | 17.92 | 0.0006 | 0.83 | 0.43 |
| | Year*Week | 6 | 14.42 | 0.001 | 0.83 | 0.43 |
| Cyprinidae | Year | 1 | 9.79 | 0.01 | 0.07 | 0.79 |
| | Week | 6 | 2.76 | 0.07 | 15.53 | 0.00 |
| | Year*Week | 6 | 2.57 | 0.09 | 10.56 | 0.00 |
| Ictaluridae | Year | 1 | 15.53 | 0.004 | 4.23 | 0.05 |
| | Week | 6 | 7.21 | 0.003 | 0.96 | 0.42 |
| | Year*Week | 6 | 7.21 | 0.003 | 1.79 | 0.16 |
| Percidae | Year | 1 | 0.07 | 0.80 | 4.32 | 0.05 |
| | Week | 6 | 0.56 | 0.66 | 3.58 | 0.03 |
| | Year*Week | 6 | 1.57 | 0.23 | 3.76 | 0.57 |
| Salmonidae | Year | 1 | 4.7 | 0.007 [a] | 2.2 | 0.15 |
| | Week | 6 | 2.4 | 0.16 [a] | 3.69 | 0.02 |
| | Year*Week | 6 | 2.4 | 0.16 [a] | 2.49 | 0.08 |

[a] Assumption of sphericity was met. The Huynh–Feldt correction factor for adjusted *p* values was not used.

The nearshore beach fish assemblage in 1993 was dominated by cyprinid species (42% of catch), alewife (31% of catch), lake whitefish (*Coregonus clupeaformis*) (11% of catch), and salmonid species (3% of catch) [35] (Table 2, Figure 3a). In spring 2012, the fish assemblages were dominated by cyprinid species (83% of catch), round goby (12% of catch), and percid species (3% of catch) (Table 2, Figure 3b). Chinook salmon, lake whitefish, and rainbow smelt were absent in 2012. Bluntnose minnow (*Pimephales notatus*), round goby, and white perch (*Morone americana*) were abundant in 2012 but absent in 1993 (Table 2, Figure 3a,b). Proportions of catch alewife, rainbow smelt, spottail shiner, trout perch, centrarchidae, and cyprinidae were higher in 1993 than in 2012. Proportion of banded killifish (*Fundulus diaphanous*), bluntnose minnow, emerald shiner, mimic shiner, round goby, yellow perch, catostomidae, and ictaluridae were higher in 2012 than 1993 (Table 2, Figure 3a,b).

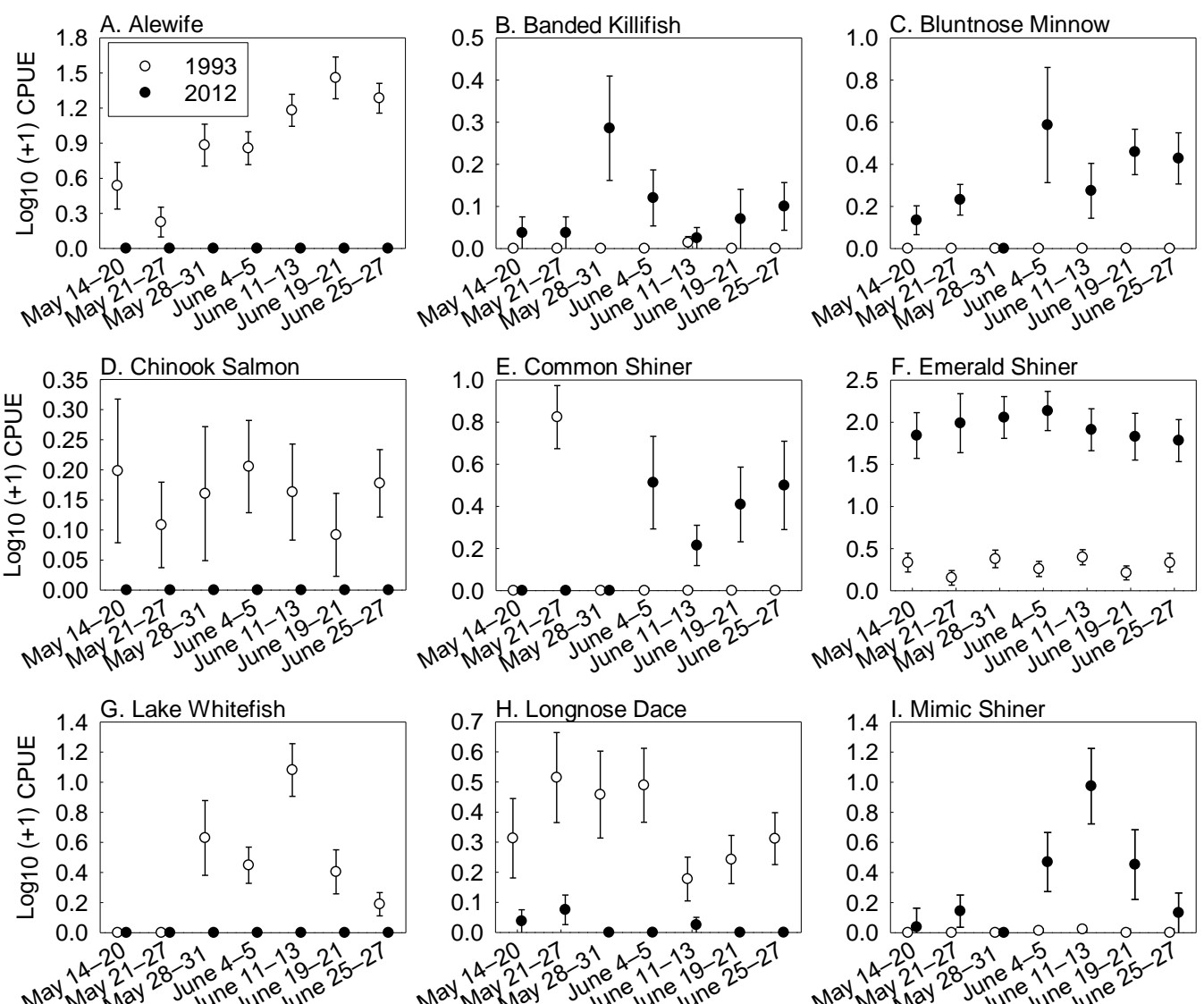

**Figure 2.** *Cont.*

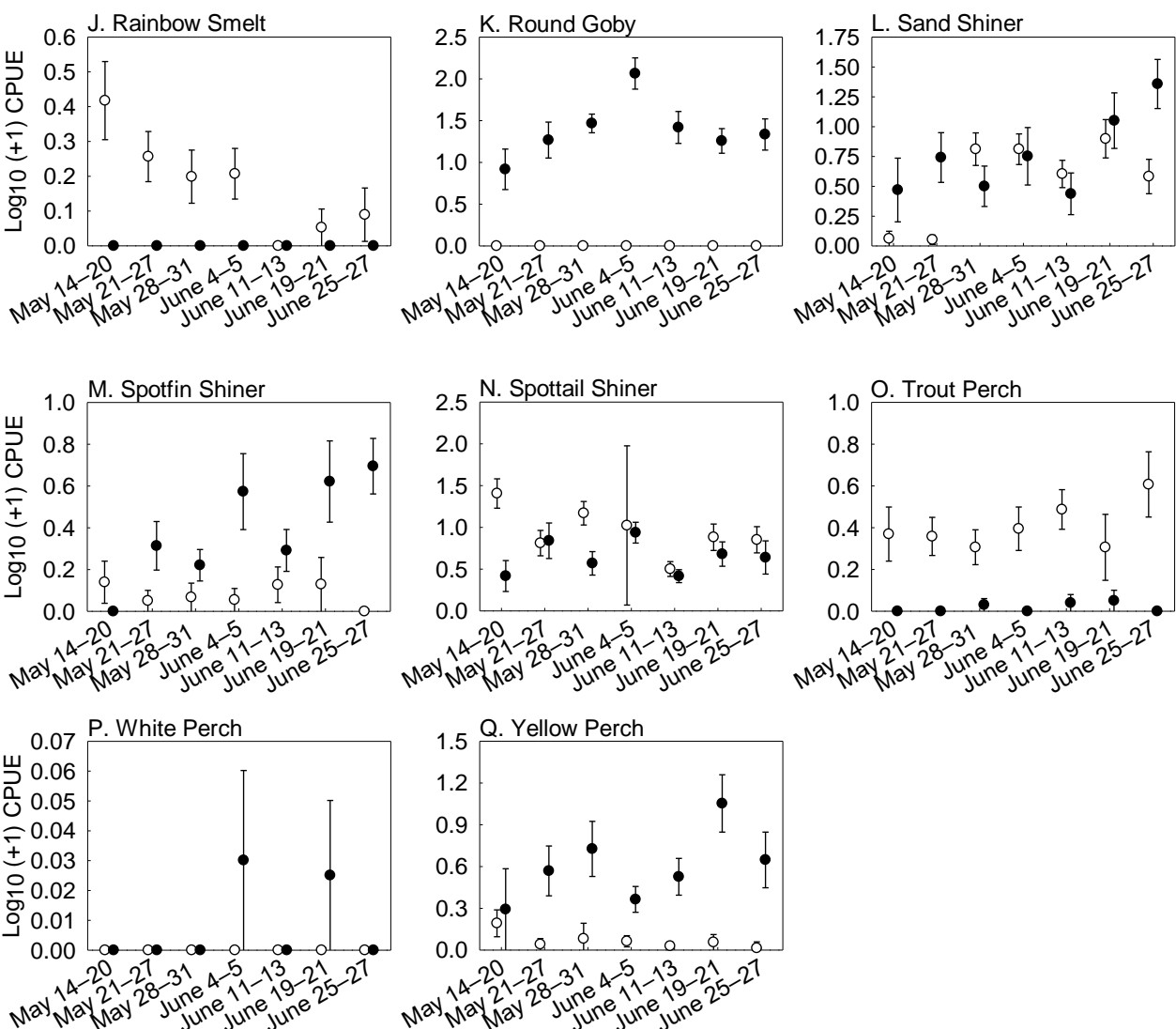

**Figure 2.** (**A–Q**) Mean (±SE) log₁₀ (+1) catch per unit effort (CPUE; #/seine haul) of the dominant species of nearshore beach fish assemblages during the spring 1993 (open circles) and spring 2012 (closed circles). Fish were collected at sites along the Michigan shoreline of western Lake Huron (see Figure 1 for sampling locations).

**Table 2.** Number of the dominant fish species that comprised 1% of the total catch in 1993 and 2012 and were present at four or more sites or within a family (not including dominant species) captured in spring 1993 and 2012. Thermal guild (cold, cool, or warm) preferred temperature, and final temperature preferendum is given for each dominant fish species or family [37,38]. Species chosen to represent the families comprised at least 50% of the catch in at least one of the sampling years. Fish were collected by seining at sites along the Michigan shoreline of western Lake Huron (see Figure 1 for sampling locations).

| Species or Family | Number Captured | | Relative Abundance (%) | | Thermal Guild [37] | Preferred Temperature [37] | Final Temperature Preferendum [38] |
|---|---|---|---|---|---|---|---|
| | **1993** | **2012** | **1993** | **2012** | | | |
| Alewife (*Alosa pseudoharengus*) | 5382 | 0 | 31.3 | 0.0 | Cold | 18.8 | 16.9 ± 4.5 |
| Bluntnose Minnow (*Pimephales notatus*) | 0 | 493 | 0.0 | 1.3 | Warm | 29 | 24.1 ± 4.5 |

**Table 2.** *Cont.*

| Species or Family | Number Captured | | Relative Abundance (%) | | Thermal Guild [37] | Preferred Temperature [37] | Final Temperature Preferendum [38] |
|---|---|---|---|---|---|---|---|
| | 1993 | 2012 | 1993 | 2012 | | | |
| Chinook salmon (*Oncorhynchus tshawytscha*) | 275 | 0 | 1.6 | 0.0 | Cold | 17.3 | 13.8 ± 2.5 |
| Common shiner (*Luxilus cornutus*) | 410 | 330 | 2.4 | 0.9 | Cool | 21.9 | 21.9 |
| Emerald shiner (*Notropis atherinoides*) | 472 | 27,197 | 2.7 | 71.8 | Cool | 22-25 | 19.3 ± 8.9 |
| Lake whitefish (*Coregonus clupeaformis*) | 2045 | 0 | 11.9 | 0.0 | Cold | 12.7 | — |
| Longnose dace (*Rhinichthys cataractae*) | 924 | 4 | 5.4 | 0.0 | Cool | 20.6 | 15.3 ± 3.6 |
| Mimic shiner (*Notropis volucellus*) | 3 | 988 | 0.0 | 2.6 | Warm | — | — |
| Rainbow smelt (*Osmerus mordax*) | 211 | 0 | 1.2 | 0.0 | Cold | 15 | 11.2 ± 3.9 |
| Round goby (*Neogobius melanostomus*) | 0 | 4613 | 0.0 | 12.2 | Cool * | — * | — * |
| Sand shiner (*Notropis stramineus*) | 1716 | 1588 | 10.0 | 4.2 | Warm | — | — |
| Spotfin shiner (*Cyprinella spiloptera*) | 358 | 436 | 2.1 | 1.2 | Warm | 29.5 | 27.5 ± 3.7 |
| Spottail shiner (*Notropis hudsonius*) | 3693 | 579 | 21.5 | 1.5 | Cold/cool | 14.3 | 16.6 ± 3.7 |
| Trout perch (*Percopsis omiscomaycus*) | 880 | 6 | 5.1 | 0.0 | Cold | 15-16 | 13.4 ± 3.5 |
| Yellow perch (*Perca flavescens*) | 69 | 1035 | 0.4 | 2.7 | Cool | 21.4 | 17.6 ± 6.0 |
| Catastomidae [1] | 95 | 386 | 0.6 | 1.0 | Cool | 22.4 | 23.4 |
| Centrarchidae [2] | 13 | 12 | 0.1 | 0.0 | Warm | 30.3 | 25.0 ± 6.0 |
| Cottidae [3] | 87 | 0 | 0.5 | 0.0 | Cold | 10; 13 | 11.0 |
| Cyprinidae [4] | 210 | 3 | 1.2 | 0.0 | Warm | 29.7 | 27.7 |
| Esocidae [5] | 2 | 0 | 0.0 | 0.0 | Cool | 22.5 | 20.7 ± 2.5 |
| Fundulidae [6] | 1 | 19 | 0.0 | 0.1 | Cool | 21 | 23.0 ± 5.0 |
| Gasterosteidae [7] | 167 | 0 | 1.0 | 0.0 | Cold | 9–10, 15–16 [1] | 16.5 ± 4.4 |
| Ictaluridae [8] | 2 | 155 | 0.0 | 0.4 | Warm | 24.9; 27.3 | 26.2 ± 2.6 |
| Lepisosteidae [9] | 4 | 0 | 0.0 | 0.0 | Warm | 33.1 | 27.4 ± 6.6 |
| Moronidae [10] | 1 | 2 | 0.0 | 0.0 | Warm | 32.0 | 29.8 |
| Percidae [11] | 34 | 27 | 0.2 | 0.1 | Cool | 22.8 | 22.8 |
| Petromyzontidae [12] | 6 | 0 | 0.0 | 0.0 | Cold | 6-15 | 10.3 ± 3.4 |
| Salmonidae [13] | 143 | 1 | 0.8 | 0.0 | Cold/cool | 21.1 | 15.7 ± 2.1 |
| Sciaenidae [14] | 3 | 0 | 0.0 | 0.0 | Warm | 26.5 | 24.6 ± 6.2 |
| Umbridae [15] | 0 | 2 | 0.0 | 0.0 | Cool/warm | — | — |
| Total | 17,206 | 37,876 | | | | | |

* Round goby listed as 'cool' without a preferred temperature Cooker et al., 2001 [37] and not identified by Hasnain et al., 2013 [38], but Kornis et al. (2012) [39] report an energetic optimum at 26 °C more consistent with a "warm" water designation. [1] Catastomidae data reported for white sucker (*Catostomus commersonii*). [2] Centrarchidae data reported for smallmouth bass (*Micropterus dolomieui*). [3] Cottidae data reported for slimy sucker (*Cottus cognatus*). [4] Cyprinidae data reported for carp (*Cyprinus carpio*). [5] Esocidae data reported for northern pike (*Esox lucius*). [6] Fundulidae data reported for banded killifish (*Fundulus diaphanous*), the only species in this family captured in this study. [7] Gasterosteidae data reported for ninespine stickleback (*Pungitius pungitius*). Cooker et al., 2001 [37] report a bimodal temperature preferendum. [8] Ictaluridae data reported for brown bullhead (*Ameiurus nebulosus*). [9] Lepisosteidae data reported for longnose gar (*Lepososteus osseus*). [10] Moronidae data reported for white perch (*Morone americana*). [11] Percidae data reported for Johnny darter (*Etheostoma nigrum*). [12] Petromyzonidae data reported for sea lamprey (*Petromyzon marinus*), the only species in this family captured in this study. [13] Salmonidae data reported for brown trout (*Salmo trutta*). [14] Sciaenidae data reported for freshwater drum (*Aplodinotus grunniens*), the only species in this family captured in this study. [15] Umbridae data reported for central mudminnow (*Umbra limi*), the only species in this family captured in this study.

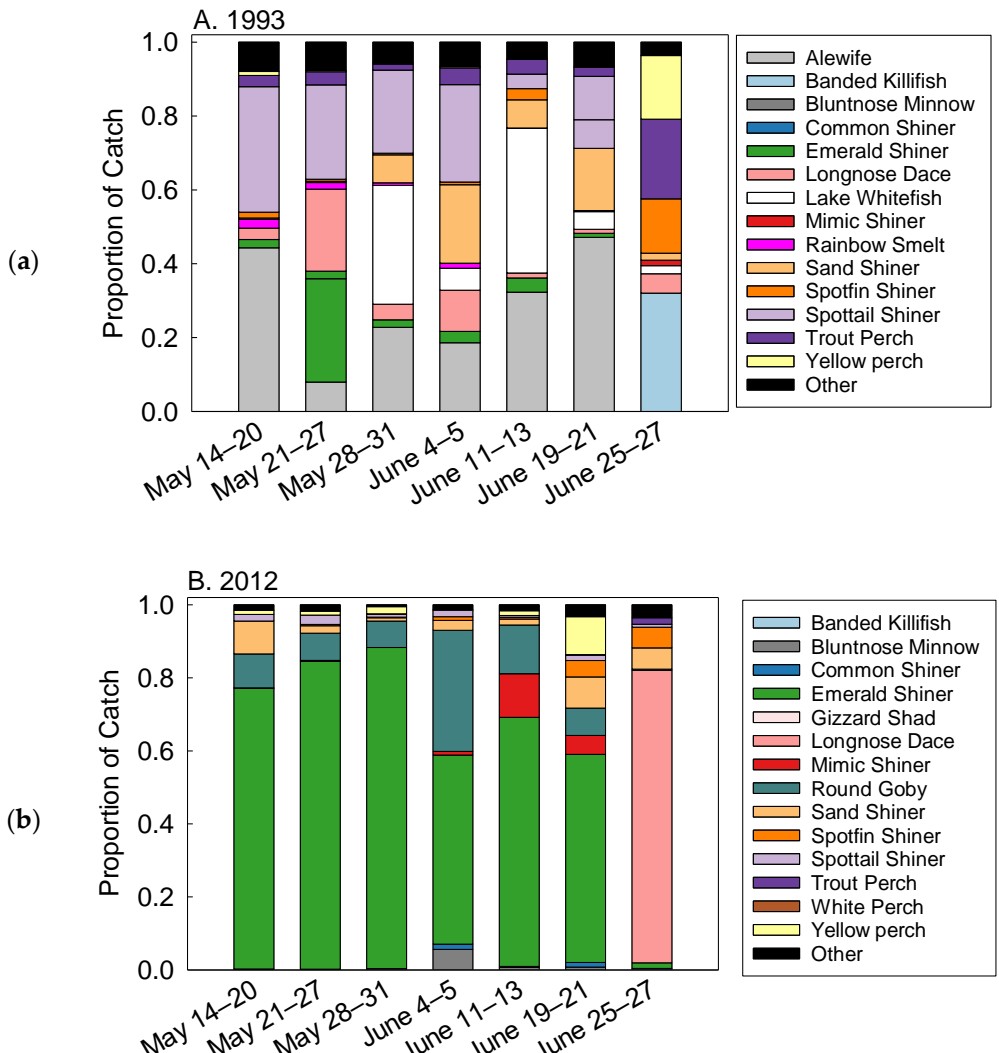

**Figure 3.** (**a**) Proportion of catch by number of dominant species and families in spring 1993 (**a**) and 2012 (**b**). Fish were collected by seining at sites along the Michigan shoreline of western Lake Huron (see Figure 1 for sampling locations). Chinook salmon, lake whitefish, and rainbow smelt were absent in 2012, while bluntnose minnow (*Pimephales notatus*), round goby, and white perch (*Morone americana*) were absent in 1993. The "other" include species that were relatively rare and were enumerated at the family level including species from Catastomidae, Centrarchidae, Cyprinidae, Ictaluridae, Percidae, and Salmonidae other than the abundant species listed above.

NMDS analyses comparing 1993 and 2012 indicated that three-dimensional ordinations were optimal. Two axes collectively explained 55.7% of the variation between these two time periods (Figure 4). The 2012 nearshore beach fish assemblages were clearly distinguished from the 1993 assemblages (MRPP: $A = 0.12$, $p < 0.001$; Figure 4). Alewife, brook stickleback (*Culaea inconstans*), brown trout, common carp, Chinook salmon, johnny darter (*Etheostoma nigrum*), longnose dace, ninespine stickleback (*Pungitius pungitius*), rainbow trout, sculpin spp., sea lamprey, rainbow smelt, and spottail shiner were correlated with axis 1 ($p < 0.05$, $n = 84$; Table 3). Alewife, brassy minnow (*Hybognathus hankinsoni*), brown trout, brook stickleback, emerald shiner, lake chub (*Couesius plumbeus*), longnose dace, lake whitefish, ninespine stickleback, rainbow smelt, rainbow trout, and sand shiner were correlated with axis 2 (Table 3).

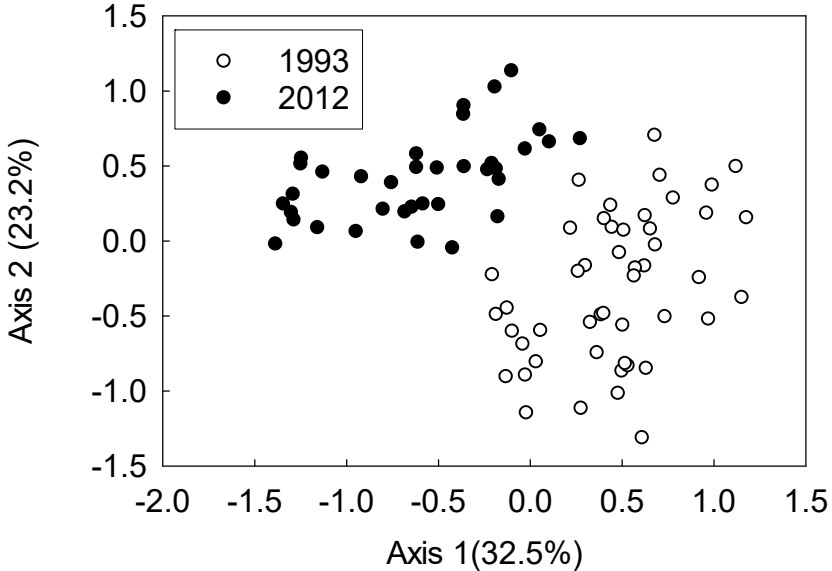

**Figure 4.** Results of NMDS ordination of nearshore beach fish assemblages over time in spring 1993 (open circles) and spring 2012 (closed circles). Each circle represents a sampling event (i.e., seine haul/date/site) during the time period. Axis 1 explains 32.5% of the variation, while Axis 2 explains 23.2% of the variation among sampling events. Axes are defined by correlation values which are presented in Table 3.

**Table 3.** Pearson and Kendall correlation r values for fish species that were significantly correlated with the NMDS ordination of nearshore beach fish assemblages in spring 1993 and 2012. Axis 1 explains 32.5% of the variation, while Axis 2 explains 23.2% of the variation among sampling events (Figure 2).

| Species | Axis 1 | Axis 2 |
|---|---|---|
| Alewife (*Alosa pseudoharengus*) | 0.31 | 0.46 |
| Brassy minnow (*Hybognathus hankinsoni*) | - | 0.28 |
| Brook stickleback (*Culaea inconstans*) | 0.23 | 0.27 |
| Brown trout (*Salmo trutta*) | 0.23 | 0.30 |
| Chinook salmon (*Oncorhynchus tshawytscha*) | 0.24 | - |
| Common carp (*Cyprinus carpio*) | 0.27 | - |
| Emerald shiner (*Notropis atherinoides*) | - | 0.38 |
| Johnny darter (*Etheostoma nigrum*) | 0.25 | - |
| Lake chub (*Couesius plumbeus*) | - | 0.29 |
| Lake whitefish (*Coregonus clupeaformis*) | - | 0.26 |
| Longnose dace (*Rhinichthys cataractae*) | 0.25 | 0.25 |
| Ninespine stickleback (*Pungitius pungitius*) | 0.29 | 0.30 |
| Rainbow smelt (*Osmerus mordax*) | 0.26 | 0.25 |
| Rainbow trout (*Oncorhynchus mykiss*) | 0.25 | 0.3 |
| Sand shiner (*Notropis stramineus*) | - | 0.35 |
| *Sculpin* spp. | 0.23 | - |
| Sea lamprey (*Petromyzon marinus*) | 0.25 | - |
| Spottail shiner (*Notropis hudsonius*) | 0.37 | - |

Indicator species analysis demonstrated similar changes. In 1993, alewife (IV (Indicator value) = 81.2, *p* < 0.001), common carp (IV = 54.2, *p* < 0.001), longnose dace (*Rhinichthys cataractae*) (IV = 60.1, *p* < 0.001), and spottail shiner (*Notropis hudsonius*) (IV = 79.3, *p* < 0.001) were indicator species (important species that define/contribute substantially to assemblage structure). In 2012, bluntnose minnow (IV = 72.2, *p* < 0.001), emerald shiner (*Notropis atherinoides*) (IV = 98.7, *p* < 0.001), round goby (IV = 100, *p* < 0.001), white sucker (*Catostomus commersonii*) (IV = 67.9, *p* < 0.001), and yellow perch (*Perca flavescens*) (IV = 84.7, *p* < 0.001) were indicator species.

Regionally, mean monthly water temperature from the offshore buoy was highly variable in both May and June (Figure 5a). May temperatures ranged between 1.83 °C (1993) and 7.54 °C (2012), and May temperatures increased significantly over the time period at a rate of 0.14 °C/year ($r^2 = 0.31$, $p = 0.01$). Similarly, June mean monthly temperatures ranged between 4.78 °C (1996) and 14.03 °C (2012) and increased at a rate of 0.19 °C/year ($r^2 = 0.21$, $p = 0.04$). Within a sampling period in a given year, water temperatures increased over the sampling period (1993: mean = 14.4 °C, rate = 0.17 °C/day, $r^2 = 0.60$, $p < 0.0001$; 2012: mean = 16.6 °C, rate = 0.23 °C/day $r^2 = 0.68$; $p < 0.001$). The increase in temperature (i.e., slope) through the sampling season was higher in 2012 ($F_{1,211} = 10.40$, $p = 0.002$, Figure 5b).

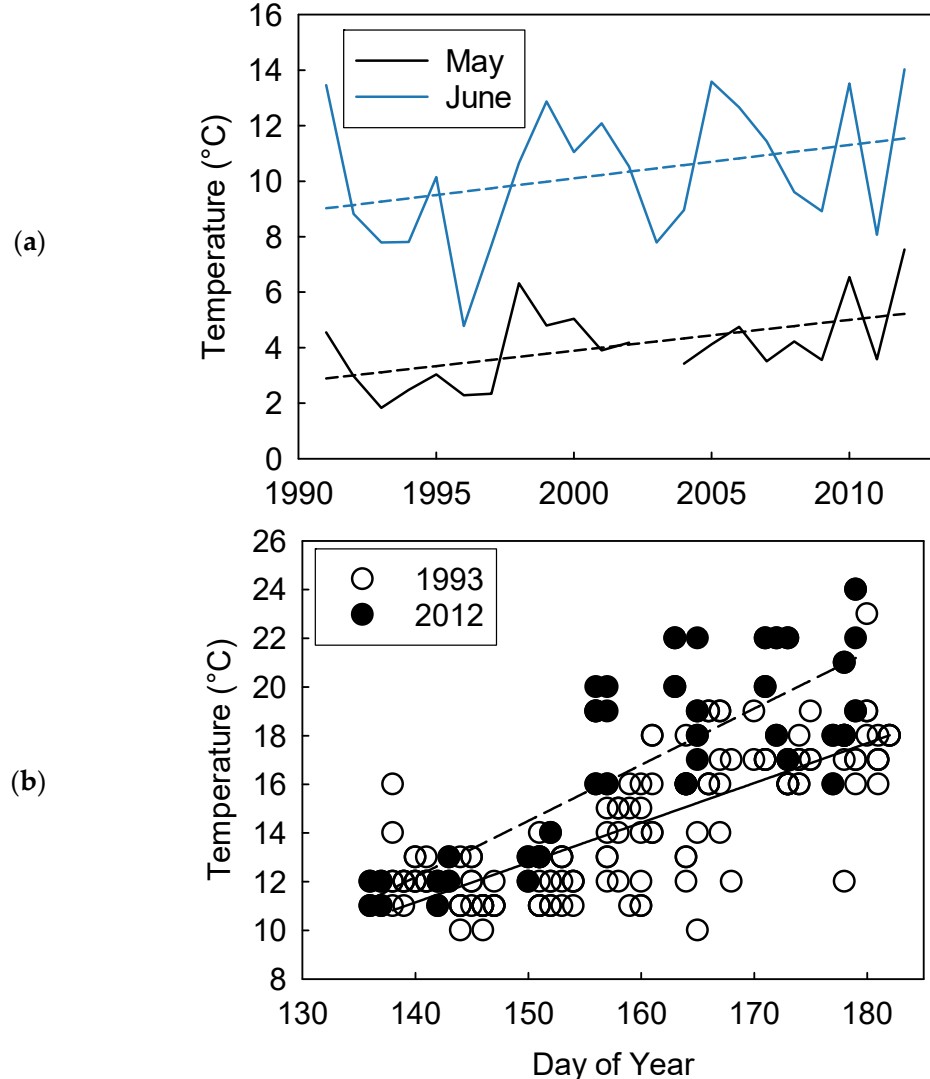

**Figure 5.** (**a**) Mean monthly water temperatures (°C) during May (solid black line) and June (dashed black line) from 1991–2012 derived from NOAA buoy 45008 representing western Lake Huron. Fitted lines for monthly water temperatures (°C) over time are given for May (solid grey line) and June (dashed grey line) 1991–2012. (**b**) Mean daily water temperature (°C) during May and June in 1993 (open circles) and 2012 (closed circles). Fitted lines for daily water temperatures (°C) over time in 1993 (solid line) and 2012 (dashed line).

Concurrent with the change in water temperature was a distinct shift from dominance of cold- and cool-water species to cool- and warm-water species. The thermal guild composition of the fish changed through a 99.8% reduction in cold-water species, a 1079.8% increase in cool-water species, and 221.0% increase in warm-water species (Table 4). The number of

dominant fish species assigned to the "cold" thermal guild captured decreased between the sampling years (alewife = −100%, Chinook salmon = −100%, rainbow smelt = −100%, spottail shiner = −84.3%, and trout perch = −99.3%) (Table 2). Mean preferred temperature and/or final temperature preferendum ranged 11.2–18.8 °C for the species belonging to the cold thermal guild (Table 2). Conversely, many species or families belonging to the cool or warm thermal guild increased or did not dramatically decrease between sampling periods (Table 2).

**Table 4.** Catch per seine haul of cold-, cool-, and warm-water fish during each sampling period derived from Table 2. Species and families designated in Table 2 as cold/cool or cool/warm were considered cool in this summarization (this includes round goby, spottail shiner, Salmonidae, and Umbridae). There were a total of 147 seine hauls in 1993 and 73 in 2012. The final column shows percent change from 1993 to 2012.

| | Catch per Seine Haul | | |
|---|---|---|---|
| **Thermal Guild** | **1993** | **2012** | **% Change** |
| Cold | 61.6 | 0.1 | −99.8 |
| Cool | 39.7 | 468.4 | 1079.8 |
| Warm | 15.7 | 50.4 | 221.0 |
| Total | 117.0 | 518.8 | 343.5 |

## 4. Discussion

We observed a dramatic regime shift along with an increase in abundance and decrease in species richness in the nearshore beach fish assemblage in western Lake Huron between 1993 and 2012. We estimated a 343% increase in CPUE, but a simultaneous 38% decrease in fish species richness. The most noticeable difference was the loss of the historically dominant cold- and cool-water species such as alewife, Chinook salmon, lake whitefish, and rainbow smelt, many of which are non-native, and were replaced by 2012 by native cool- and warm-water species such as cyprinids (especially emerald shiner), yellow perch, and white sucker along with the non-native round goby. The regime shift in the beach fish assemblage appears related to changes in Lake Huron as a whole, reflecting food web changes precipitated by the invasion of two keystone species, dreissenid mussels and round goby [6,13,40–43]. Concomitant with the food web changes has been a slow but steady warming of nearshore surface waters that may be related to the transition toward cool- and warm-water species.

The NOAA buoy in western Lake Huron, offshore from Oscoda, MI, documented an average annual increase of May and June mean water temperature of 0.14 °C and 0.19 °C, respectively from 1993 to 2012. Similarly, the rate of within season warming in the shallow nearshore waters from May through June increased from 0.17 °C/day in 1993 to 0.23 °C/day in 2012. Warmer temperatures as well as daily rates of increase will affect the suitability of coastal habitats for cold- and cool-water species nursery needs [30,44]. Because our data represent two end points, i.e., 1993 and 2012, we lack information on the timing of changes in species abundance relative to warming and invasions so we cannot definitively say whether the regime shift was driven by temperature or invasive species, although it is likely that the warming environmental conditions facilitated the species changes.

Changes in the offshore food web and specifically declines in salmonid, alewife, and lake whitefish populations coincided with the timing of the non-native dreissenid mussel and round goby invasions in Lake Huron [6,13,40–43,45,46]. Similar food web changes associated with dreissenid mussels and round goby have been documented in Lakes Michigan and Ontario. The root of the complex food web changes appears to be related to declines in *Diporeia*, an important pelagic and deep benthic invertebrate food source, impacted by competition with dreissenid mussels [24,43]. The loss of *Diporeia* as prey contributed to the decline in alewife and lake whitefish populations [7,25,46,47]. As alewife numbers declined, the Chinook salmon fishery collapsed [21,27]. All of these

species utilized nearshore beaches as nursery habitats. It is also worth noting that, although their abundances are lower than historic levels, alewife, Chinook salmon, rainbow smelt, and lake whitefish remain relatively common in Lake Huron. Their continued persistence along with the observation that they no longer utilize nearshore beaches for nursery habitat suggests they have shifted habitat use during the nursery stage, likely to cooler offshore waters. This habitat shift could also be related to their population decline as predators are likely more abundant and prey less available (e.g., loss of spring plankton bloom) [48].

The nearshore area serves as nursery habitat dominated by small and young fish, and the use of this habitat changes with water temperatures [6,49]. Approximately 80% of all Great Lakes fish species use nearshore habitat at some point during their life cycles [34,49,50]. Direct changes in habitat suitability associated with seasonal warming interact with complex direct and indirect onshore–offshore, benthic–pelagic, and seasonal food web linkages to continue to change Lake food web structure. Invasive species such as dreissenid mussels and round goby continue to alter energy flow from offshore to nearshore areas [14] and these changes will continue as habitats and species composition adapt and evolve to ever changing thermal regimes [6,30]. The loss of cold- and many cool-water species such as lake whitefish and other salmonid species underscores the substantial biotic and abiotic changes that have occurred over the last few decades. Although there have been many factors that can influence fish assemblages, the fish assemblages in the Great Lakes are always changing with the intentional and unintentional introduction of non-native species, consequently changing Lake Huron's food web structure forever [9,10]. Continued sampling becomes increasingly important in identifying long-term trends across the Great Lakes and the effects on the rapidly changing ecosystem.

**Author Contributions:** All authors have made substantial contribution to this manuscript (outline ahead), have approved the final draft of the manuscript, and agree to be personally accountable for the integrity of the work. Specific contributions: conceptualization (J.B., T.G. and J.J.), methods/approach (J.B., T.G. and J.J.), field work (J.B., T.G., B.M. and J.J.), statistical approaches and analysis (J.B., T.G. and B.M.), writing of the original draft (J.B.), subsequent editing and revision (J.B., T.G. and B.M.), project administration and funding acquisition (T.G.). All authors have read and agreed to the published version of the manuscript.

**Funding:** This work was funded by the Michigan Department of Natural Resources Game and Fish Fund, Central Michigan University and the U.S. Environmental Protection Agency under the Coordinated Science and Monitoring Initiative, Lake Huron in 2012. This material is based upon work that is partially supported by the National Institute of Food and Agriculture, U.S. Department of Agriculture, McIntire Stennis project under 1026001.

**Institutional Review Board Statement:** The study was conducted according to the guidelines of the Declaration of Helsinki and approved by the Institutional Animal Care and Use Committee of Central Michigan University (IACUC # 12-08).

**Data Availability Statement:** Fish data are available upon request, historic (1993) data were collected by U.S. state and federal agencies but predate public online data repositories and the current data (2012) are held by Central Michigan University without an available public online interface. The water temperature data are openly available through NOAA https://www.ndbc.noaa.gov/station_history.php?station=45008.

**Acknowledgments:** The authors would like to thank Andrew Briggs, Mikel Vredevoogd, Eric Calabro, Lindsey Adams, Cassie Dresser, Courtney Higgins, and Sara Thoma for assistance with field work and fish identification. We also thank the three anonymous reviewers for providing comments for improvement of this work. The findings and conclusions in this article are those of the authors and do not necessarily represent the views of the U.S. Fish and Wildlife Service. Scientific Article No: 3441 of the West Virginia Agricultural and Forestry Experiment Station, Morgantown, WV.

**Conflicts of Interest:** The authors declare no conflict of interest.

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
