# Peer review of "Invasive Species Appearance and Climate Change Correspond with Dramatic Regime Shift in Thermal Guild Composition of Lake Huron Beach Fish Assemblages"

_fishes, doi:10.3390/fishes7050263_

Round 1

Reviewer 1 Report

This manuscript is a study on the temporal change of fish assemblages that corresponds with increasing temperatures and new invasions in the area. It goes along with the Journal scope.

The manuscript reads very well, and it is very clear and interesting.

I have mainly minor comments, generally related to style and format. The only major comment that I see this manuscript is lacking, is a more complete Discussion section. I find it too short and several points that are missing and that if added, they could certainly bonify the work. Many subjects can be discussed to make this section a little bit more comprehensive. A few ideas of subjects that can be expanded:

-any studies or results that can be compared related to increase of temperature and invasion in other regions of the lakes of Great Lakes?

-what are the expected changes for this lake under global warming scenario? More warm species (even the cool species may tend to disappear)

-Which other invasive species could play a bigger role?

-What changes in the habitat could also play an important role?

-Are there any other places in the world that have seen this change in the assemblage after the increase in temperature and invasive species?

-What are the consequences of these changes in the assemblage? Not only ecologically, but a mention on economic impact or sport fishing impacts could easily be mentioned.   

Bellow, a few specific comments:

1.    Figure 1: make the region more visible, showing the other Great lakes and a little bit more of information on where Michigan is situated in the States and frontier with Canada.

2.    Line 121: briefly explain what MS-222 is.

3.    Lines 129, 164, 325, 333: Extra space between words.

4.    Line 152: I believe it should say “was used” instead of “were used”.

5.    Lines 154-158: repetitive from the previous paragraph? If not, I don’t understand the difference and it makes it difficult to follow.

6.    Line 160: explain the words contained in the acronym SAS

7.    Figures 2 and 3: suggest adding info in the legend. More visual is better than reading to interpret. Instead of writing open and closed circles, they should be visually added in the legend (rather than in the caption).

8.    Figures 3 and 4: weeks in the graphs could be added as number and in the figure caption explain the exact dates.

9.    Figure 4: Is there a way of using other symbology? Sometimes difficult to follow. Add the year in the figure so it's easy at a first glance to see what it is the difference between both graphs

10.  Table 2 and Figure 2: the Axes, do they represent something in particular? If yes, they should be explained. If not, just a quick reminder how this should be interpreted.

11.  Figure 5: also, add in visual legend what the color of the lines represent.

12.  Table 4: really interesting results!

13.  Line 359: “food” instead of “feed”.

References: Check that some references are not included (example: reference 33 and 38)

Author Response

Reviewer 1.  Comments and Suggestions for Authors

This manuscript is a study on the temporal change of fish assemblages that corresponds with increasing temperatures and new invasions in the area. It goes along with the Journal scope.

The manuscript reads very well, and it is very clear and interesting.

I have mainly minor comments, generally related to style and format. The only major comment that I see this manuscript is lacking, is a more complete Discussion section. I find it too short and several points that are missing and that if added, they could certainly bonify the work. Many subjects can be discussed to make this section a little bit more comprehensive. A few ideas of subjects that can be expanded:

-any studies or results that can be compared related to increase of temperature and invasion in other regions of the lakes of Great Lakes?

-what are the expected changes for this lake under global warming scenario? More warm species (even the cool species may tend to disappear)

-Which other invasive species could play a bigger role?

-What changes in the habitat could also play an important role?

-Are there any other places in the world that have seen this change in the assemblage after the increase in temperature and invasive species?

-What are the consequences of these changes in the assemblage? Not only ecologically, but a mention on economic impact or sport fishing impacts could easily be mentioned.   

These are all excellent questions, we did not however, make extensive changes to the discussion.  We feel that we do refer to the other Great Lakes that it is largely understood that similar food web effects, particularly of dreissenids and round goby have occurred in the other lakes, but we do specifically note Lakes Michigan and Ontario.  We want to avoid conjecture or making comments that extend too far beyond that which our data support, e.g. expected changes under future warming scenarios we feel would be largely conjecture, but the data we present clearly allows readers to form there own thoughts.  Relative to the time period and beach habitat we certainly feel that dreissenids and round goby are the most important invasive species, though there are certainly secondary players, none that match the impacts dreissenids and round goby have had.  We could make some speculations about water clarity (i.e., the habitat changes suggestion), but we are not providing historic or present data on how water clarity has changed, though we know and it is widely known that it has increased tremendously, largely as an artifact of the dreissenids.  Similarly, we feel discussion of economic impacts is beyond the scope of the data and would require a degree of conjecture, again, readers can certainly infer such from the data and outcomes.

Bellow, a few specific comments:  We made all the suggested changes below and are thankful to reviewers for these helpful comments.  We offer some additional thoughts where relevant below.

  1. Figure 1: make the region more visible, showing the other Great lakes and a little bit more of information on where Michigan is situated in the States and frontier with Canada.
  2. Line 121: briefly explain what MS-222 is.
  3. Lines 129, 164, 325, 333: Extra space between words. In some cases, this may be a function of the formatting  of the ms in the column justified form.
  4. Line 152: I believe it should say “was used” instead of “were used”.
  5. Lines 154-158: repetitive from the previous paragraph? If not, I don’t understand the difference and it makes it difficult to follow.

Yes, this is redundant but the first series refers to abundance and the second refers to proportions, we have made an effort to revise to make it more readable.

  1. Line 160: explain the words contained in the acronym SAS

The company started in the 1970s as “Statistical Analyses Software” but only goes by SAS now, we added a link to the company’s website.

  1. Figures 2 and 3: suggest adding info in the legend. More visual is better than reading to interpret. Instead of writing open and closed circles, they should be visually added in the legend (rather than in the caption).

We realized that some of the figures were out of place and one (now Figure 4) had an (a) and (b), but the (b) plot should have been edited out in an earlier draft.  We have revised the order of the figures and tables to align with the text.

  1. Figures 3 and 4: weeks in the graphs could be added as number and in the figure caption explain the exact dates.

We did not make this change for a couple reasons.  First, we feel that the dates add important information to make clear these were late spring and summer samples.  Second, many of the other comments on the figures aimed to make them more readable and more easily understood without referring to the caption, and we agree, those are all valuable suggestions, but this seemed to work against that concept by forcing readers to refer to the caption to understand the timing, leaving the dates seems more consistent with the overall gist of the comments and improves readability and understanding.

  1. Figure 4: Is there a way of using other symbology? Sometimes difficult to follow. Add the year in the figure so it's easy at a first glance to see what it is the difference between both graphs

Enthusiastically agree!  This figure was revised by using simple color instead of symbols and we also based on another reviewer removed the families and included them as “other” (but columns still sum to 1.0), which simplifies the columns and legend.

  1. Table 2 and Figure 2: the Axes, do they represent something in particular? If yes, they should be explained. If not, just a quick reminder how this should be interpreted.
  2. Figure 5: also, add in visual legend what the color of the lines represent.
  3. Table 4: really interesting results! Agreed
  4. Line 359: “food” instead of “feed”.

References: Check that some references are not included (example: reference 33 and 38)

Reviewer 2 Report

The article compares the abundance and fish species diversity in catches made in the same areas of the coast of Lake Huron and in the same way, at the same time of the year and day, but with an interval of 20 years. The comparison shows that the fish catches has become higher, but the diversity of fish has decreased, and the ratio of fish species has also changed. The authors attribute this to global climate change and the penetration of alien species into the lake. The article is well written, easy to read, the material is processed by adequate statistics and comprehensively analyzed.

I have the following comments, which are not critical, but require attention:

Line 2: in the title of the article, penetration/appearance should be inserted after “Invasive species ….”

Introduction: it is strange that the authors do not mention and do not take into account the entry into the Lake Huron of Eurasian ruffe Gymnocephalus cernua, attention should be paid to the possible influence of this fish on the processes under consideration.

Lines 19 and 177-118: in the article, the time of day when fish were sampled is very vaguely indicated - nighttime/after sunset. The time of fishing should be more clearly defined, since illumination strongly affects the behavior of fish and the catchability of fishing gear.

Lines 40-41: it seems to me that one thing should be left in the text - either “non-40 native” or “invasive”

Lines 85-86: the phrase “We hypothesized that fish diversity is higher in the recent time….” confuses and disorients the reader, because the results showed that the diversity of fish, in contrast, has become lower but not higher.

Line 100: in the figure it is necessary to show where the water is and where the land is, at least by color.

Results:

- I have not found a complete list of fish species that were present in the catches in 1993 and 2012. This information should be included in the article.

- there is a clear discrepancy (confusion) in the number of "dominant species" given in tables and figures: in tables 3 and 2 there are 16 and 18 such species, in figures 3 and 4 - 17 and 16. How many of them were real? And how many non-dominant fish species were there?

- Figures 4a and 4b are obviously overloaded with information, information about families should be deleted from them, or such information should be presented in new figure.

Author Response

Reviewer 2: The article compares the abundance and fish species diversity in catches made in the same areas of the coast of Lake Huron and in the same way, at the same time of the year and day, but with an interval of 20 years. The comparison shows that the fish catches has become higher, but the diversity of fish has decreased, and the ratio of fish species has also changed. The authors attribute this to global climate change and the penetration of alien species into the lake. The article is well written, easy to read, the material is processed by adequate statistics and comprehensively analyzed.

Again, we have made all the recommended changes below, but offer some additional comment as relevant.

I have the following comments, which are not critical, but require attention:

Line 2: in the title of the article, penetration/appearance should be inserted after “Invasive species ….”

Introduction: it is strange that the authors do not mention and do not take into account the entry into the Lake Huron of Eurasian ruffe Gymnocephalus cernua, attention should be paid to the possible influence of this fish on the processes under consideration.

Ruffe have been in the Great Lakes for quite a long time now, but have not had widespread effects and did not show up in either of the beach community surveys, so it did not seem to have relevance here.

Lines 19 and 177-118: in the article, the time of day when fish were sampled is very vaguely indicated - nighttime/after sunset. The time of fishing should be more clearly defined, since illumination strongly affects the behavior of fish and the catchability of fishing gear.

Lines 40-41: it seems to me that one thing should be left in the text - either “non-40 native” or “invasive”

We do use both non-native and invasive to reflect the fact that there are so many highly desirable non-native fish which are in contrast to the undesirable ones, which are termed invasive.  Most of the salmonids, and even alewife, are non-native, but presently desirable, thus while we have many non-natives, not all are problematic, we reserve and use “invasive” for those species that are non-native, undesirable, and problematic.

Lines 85-86: the phrase “We hypothesized that fish diversity is higher in the recent time….” confuses and disorients the reader, because the results showed that the diversity of fish, in contrast, has become lower but not higher.

We revised the wording, for hopefully greater clarity. However, as a note of explanation, we are reporting our a priori hypothesis that we expected higher species richness because generally warmwater fish assemblages are more diverse than coldwater assemblages and added non-native/invasive species, but turns out we were wrong, we observed fewer species and greater dominance.

Line 100: in the figure it is necessary to show where the water is and where the land is, at least by color.

Results:

- I have not found a complete list of fish species that were present in the catches in 1993 and 2012. This information should be included in the article.

We attempted to clarify this.

- there is a clear discrepancy (confusion) in the number of "dominant species" given in tables and figures: in tables 3 and 2 there are 16 and 18 such species, in figures 3 and 4 - 17 and 16. How many of them were real? And how many non-dominant fish species were there?

As mentioned earlier, while making the revisions suggested by all three reviewers we came to recognize that the figures and tables were out of order but have revised that through this process.  We have added notes to figure and table captions and revised text in an effort to improve clarity here.

- Figures 4a and 4b are obviously overloaded with information, information about families should be deleted from them, or such information should be presented in new figure.

This is consistent with comments from reviewer one, we have removed the families and added them to “other” so that the columns (proportions) still sum to 1.0.

Reviewer 3 Report

Review

Paper title: Invasive species and climate change correspond with dramatic regime shift in thermal guild composition of beach fish assemblages.

The authors conducted a field study to catch fish and compare fish assemblages in Lake Huron in two different periods: one with low or zero densities of invasive species and a colder temperature regime and the other with higher invader densities and a warmer temperature regime. The authors registered a dramatic shift in the local fish assemblage (higher fish abundance but lower species richness and changes in the structure of the fish assemblage). The authors concluded that both factors – climate change and invasion of Dreissena spp. and the round goby – were responsible for the pattern observed. These results may have important implications for management and conservation in Lake Huron.

All these reasons explain the relevance of the paper by Jessica Bowser and co-authors submitted to "Fishes".

General scores.

The data presented by the authors are original and significant. The study is correctly designed and the authors used appropriate sampling methods. In general, statistical analyses are performed with good technical standards. The authors conducted careful work that may attract the attention of a wide range of specialists focused on fish biology and biological invasions.

Recommendations.

Figure 1. The authors should add coordinates and a scale bar to represent information about the real distances between the sampling sites.

References should be formatted according to Rules for Authors.

L 470, 478. Missing data.

Specific remarks.

L 15. Consider replacing “nursery habitat” with “nursery habitats”

L 22. Consider replacing “from” with “from a”

L 46. Consider replacing “1980’s” with “1980s”

L 87. Consider replacing “cold and cool water” with “cold- and cool-water”

L 108. Consider replacing “northern most” with “northernmost”

L 132. Consider replacing “p value” with “p-value”

L 152. Consider replacing “P  values  were” with “p-values  was”

L 183. Consider replacing “significant interaction” with “a significant interaction”

L 201. Consider replacing “absent 1993” with “absent in 1993”

L 202. Consider replacing “catch of” with “catch in”

L 205. Consider replacing “(Coker et al., 2001;  Hasnain et al., 2013)” with “[35, ?]” Note that Hasnain et al. 2013 is missing in the reference list! Also, revise accordingly the first line of Table 1.

L 210. Kornis et al. (2012) is missing in the reference list!

L 219. Consider replacing “Coker et al. (2001)” with “[35]”

L 241. Consider replacing “indicated. .” with “indicated.”

L 246. Consider replacing “than 2012” with “than in 2012”

L 296. Consider replacing “time 1993” with “time in 1993”

L 313. Consider replacing “2012. .” with “2012.”

L 347. Consider replacing “habitat” with “habitats”

Author Response

Reviewer 3.

The authors conducted a field study to catch fish and compare fish assemblages in Lake Huron in two different periods: one with low or zero densities of invasive species and a colder temperature regime and the other with higher invader densities and a warmer temperature regime. The authors registered a dramatic shift in the local fish assemblage (higher fish abundance but lower species richness and changes in the structure of the fish assemblage). The authors concluded that both factors – climate change and invasion of Dreissena spp. and the round goby – were responsible for the pattern observed. These results may have important implications for management and conservation in Lake Huron.

All these reasons explain the relevance of the paper by Jessica Bowser and co-authors submitted to "Fishes".

The data presented by the authors are original and significant. The study is correctly designed and the authors used appropriate sampling methods. In general, statistical analyses are performed with good technical standards. The authors conducted careful work that may attract the attention of a wide range of specialists focused on fish biology and biological invasions.

Recommendations.  All recommended changes have been made.

Figure 1. The authors should add coordinates and a scale bar to represent information about the real distances between the sampling sites.

References should be formatted according to Rules for Authors.

L 470, 478. Missing data.

Specific remarks.

L 15. Consider replacing “nursery habitat” with “nursery habitats”

L 22. Consider replacing “from” with “from a”

L 46. Consider replacing “1980’s” with “1980s”

L 87. Consider replacing “cold and cool water” with “cold- and cool-water”

L 108. Consider replacing “northern most” with “northernmost”

L 132. Consider replacing “p value” with “p-value”

L 152. Consider replacing “P  values  were” with “p-values  was”

L 183. Consider replacing “significant interaction” with “a significant interaction”

L 201. Consider replacing “absent 1993” with “absent in 1993”

L 202. Consider replacing “catch of” with “catch in”

L 205. Consider replacing “(Coker et al., 2001;  Hasnain et al., 2013)” with “[35, ?]” Note that Hasnain et al. 2013 is missing in the reference list! Also, revise accordingly the first line of Table 1.

L 210. Kornis et al. (2012) is missing in the reference list!

L 219. Consider replacing “Coker et al. (2001)” with “[35]”

L 241. Consider replacing “indicated. .” with “indicated.”

L 246. Consider replacing “than 2012” with “than in 2012”

L 296. Consider replacing “time 1993” with “time in 1993”

L 313. Consider replacing “2012. .” with “2012.”

L 347. Consider replacing “habitat” with “habitats”